# Benefit-cost analysis of raccoon rabies control in Ontario, Canada

**Stephanie A. Shwiff** [1]*, **Emily S. Acheson** [2,3,4], **Levi Altringer** [1], **Patrick A. Leighton** [2,3], **Larissa Nituch** [5], **Sarah Sykora** [1,6], **François Viard** [2,3], **Tore Buchanan** [5]

**1** USDA APHIS Wildlife Services, National Wildlife Research Center, Fort Collins, Colorado, United States of America, **2** Research Group on Epidemiology of Zoonoses and Public Health (GREZOSP), Faculty of Veterinary Medicine, University of Montréal, Saint-Hyacinthe, Canada, **3** Centre de Recherche en Santé Publique, l'Université de Montréal et du CIUSSS du Centre-Sud-de-l'île-de-Montréal (CReSP), Montréal, Canada, **4** Public Health Risk Sciences Division, National Microbiology Laboratory, Public Health Agency of Canada, Saint-Hyacinthe, Canada, **5** Wildlife Research and Monitoring Section, Ontario Ministry of Natural Resources and Forestry, Trent University, Peterborough, Canada, **6** Department of Economics, Colorado State University, Fort Collins, Colorado, United States of America

* stephanie.a.shwiff@usda.gov

## Abstract

Zoonotic diseases, particularly those originating in wildlife, pose significant public health and economic risks. Rabies, a viral zoonosis with near-100% case fatality in humans, is a prime example of such a threat, especially in regions like North America where wildlife—such as raccoons—serve as key reservoirs. This study assesses the economic efficiency of Ontario, Canada's raccoon rabies control program, which combines oral rabies vaccination (ORV), trap-vaccinate-release (TVR), and surveillance strategies. Using a spatial agent-based epidemiological model, the study estimates the benefits and costs of intervention compared to a no-intervention scenario over the period 2015–2025. Benefits were quantified as avoided public health costs, including post-exposure prophylaxis (PEP), animal testing (AT), and human exposure investigations (INVT), and converted to 2023 CAD. Results show that the intervention prevented significant economic losses, with benefit-cost ratios ranging from 1.5 to 14.16 depending on assumed rates of intervention necessity, confirming the program's cost-effectiveness. This analysis not only supports continued investment in wildlife rabies control but also provides a scalable economic framework for other zoonotic disease management programs utilizing a One Health approach.

## Author summary

In 2015, Ontario faced the largest raccoon rabies outbreak in Canadian history. The Ontario government implemented control measures to contain the outbreak. Our research explores the economic benefit of intervention strategies used by modeling three scenarios: continued control measures until 2025, ending control

the Creative Commons CC0 public domain dedication.

Data availability statement: The rabies case data used in this study are available on the Canadian Food Inspection Agency website (https://inspection.canada.ca/animal-health/terrestrial-animals/diseases/reportable/rabies/rabies-in-canada/eng/1356156989919/1356157139999) and Ministry of Natural Resources and Forestry website (https://www.ontario.ca/page/rabies-cases). Additional spread simulation data can be found at https://doi.org/10.3390/v15020528.

Funding: The author(s) received no specific funding for this work.

Competing interests: The authors have declared that no competing interests exist.

measures in 2020, and no interventions at all. By estimating the human population at risk under each scenario, we monetized the benefits of intervention. Our findings indicate that continuing control measures until 2025 would prevent the most cases of rabies, significantly reducing healthcare costs associated to post-exposure prophylaxis, animal rabies testing, and human exposure investigations. The benefit-cost analysis shows that for every dollar spent on rabies control, between $1.5 CAD and $14 CAD in healthcare costs were saved, depending on the scenario. This translates into total savings ranging from $53 million CAD to $495 million CAD by 2025. This research underscores the value of sustained intervention efforts in managing wildlife diseases and protecting public health. It highlights how proactive management cand lead to substantial economic and health benefits, reinforcing the importance of the "One Health" approach, which integrates human, animal, and environmental health strategies.

## Introduction

Research indicates that preventing the emergence of zoonotic diseases is more cost effective than disease suppression once an outbreak has occurred [1]. Wildlife have been implicated as a main driver of pathogen emergence through wildlife engagement (e.g., hunting, recreational uses and trade), agricultural expansion into wild areas, and destruction of wildlife habitat [1]. Zoonotic diseases are caused by pathogens that can be transmitted between animals and humans, and many originate in wildlife, like rabies, the zoonosis with the highest case fatality rate [2]. The development of strategies to prevent or control the spread of zoonotic diseases, like rabies, can provide a means of measuring the economic efficiency of managing zoonosis in wildlife providing a blueprint to combat the emergence of other wildlife vectored zoonoses.

Rabies, an acute and nearly always fatal, viral zoonosis unique to mammals, is primarily transmitted through saliva, or less commonly, brain or nervous system tissue [3]. Rabies prevention has taken a "One Health" approach, which applies a coordinated, collaborative, multidisciplinary, and cross-sectoral effort to address potential or existing risks that originate at the animal-human-ecosystems interface [2,4]. While no effective treatment exists for clinical rabies virus infection, efficacious vaccines are available for humans, domestic animals, and wildlife, offering a significant opportunity for prevention. Rabies presents a unique challenge due to its near 100% human case fatality rate, rapid progression, and use of post-exposure prophylaxis (PEP) as the only means of preventing clinical rabies in humans in the absence of vaccination. Economically, this interplay of factors results in the majority of rabies-related costs stemming not from treating illness, but from human deaths and preventive measures. While globally, 59,000 – 69,000 people die from mostly dog rabies annually, in North America, where rabies primarily affects wildlife (e.g., raccoons, skunks, foxes, bats), human exposure in this region often occurs through interaction with wildlife, with dogs still playing a role for human rabies exposure, either through direct interaction or via contact with rabid wildlife followed by transmission to humans [4,5].

The central goal of rabies management is to prevent transmission from animals to humans and the central role of economic analysis of rabies management is to inform the optimization of rabies prevention strategies. Success can be measured by avoided costs (e.g., reduced PEP, animal testing, livestock losses) or the number of individuals protected [6–9]. In North America, the economic burden of rabies necessitates management strategies focused largely on wildlife vaccination programs. Evaluating the economic impacts of these strategies is vital for effective control and public support [7,10]. The primary prevention strategy for human rabies in this region involves large-scale wildlife vaccination, particularly targeting key reservoir species. The economic consequences of rabies encompass medical treatment for bites, PEP costs, animal vaccination expenses, and livestock losses. Economic analyses typically report the overall disease burden, economic efficiency of oral rabies vaccination programs (ORVP), cost per life saved, PEP costs, vaccination program costs, and disability-adjusted life years (DALYs) [11,12]. Notably, PEP, because of its expense and administration in cases that may not warrant it, often constitutes the largest portion of treatment expenditures [13–17]. A standard PEP regimen involves multiple vaccine doses and rabies immunoglobulin (RIG) administration. In the U.S., the average cost per suspected human rabies exposure, including PEP, is approximately US $3,300 (2020), representing 70% of the cost per event [18]. In North America, raccoon rabies stands out among the other wildlife variants as creating the largest economic burden [10].

There is a geographical imbalance in rabies economic literature, with a concentration in low rabies-risk developed countries and a scarcity in high-risk regions where dog-mediated transmission persists, hindering the development of cost-effective prevention strategies [11,13,19,20]. The literature is dominated by geographically limited case studies and data-dependent modeling, with underrepresentation of crucial cost-benefit (CBA) and cost-effectiveness analyses (CEA) [11,13,19,20]. Although dog-mediated rabies is a research focus reflecting its burden, economic studies on other reservoirs like wildlife are limited despite their relevance, especially in regions where canine rabies is controlled, like North America. Many of the gaps in the literature can be addressed by combining modeling and CBA or CEA as well as expanding the understanding and evaluation of rabies control through alternate methodologies such as a game theoretic concept to explore the benefits of a "One Health" collaborative approach to rabies prevention [21]. Examination of the benefits and costs of rabies prevention in Ontario, Canada addresses some of the gaps in rabies economic studies by combining actual intervention costs with benefits from an epidemiological model to estimate the benefit-cost ratios of the program.

Ontario, Canada, once considered a rabies hotspot, has successfully implemented rabies control programs for the past 33 years, led by the Ontario Ministry of Natural Resources and Forestry (MNRF), effectively eliminating various rabies epizootics [22–28]. Wildlife rabies management in Ontario includes trap-vaccinate-release (TVR) and oral rabies vaccination (ORV) bait distribution. Raccoons (*Procyon lotor*), adaptable to both urban and rural environments, pose a significant public and animal health risk due to their proximity to humans in high-density urban areas [22].

In 2015 the raccoon rabies outbreak in Hamilton, Ontario, represented the largest in Canadian history [29] (Fig 1). In Ontario, the Ministry of Health provides guidance for local human rabies exposure management and oversees rabies immunization legislation. Local public health units develop contingency plans for human rabies prevention, including PEP provision and public awareness campaigns [27,29]. The Ontario Ministry of Agriculture and Rural Affairs handles domestic animal and livestock rabies investigations, while the MNRF manages wildlife rabies surveillance and control, with all associated costs borne by the Ontario government.

Following the successful containment of raccoon rabies in Ontario, a spatial agent-based model assessed the effectiveness of preventing its spread under different spatial and temporal scenarios [22]. Model predictions indicated that rabies was most likely to disappear if control measures continued until 2025 and in all the situations studied, there was a small chance that rabies would quickly die out on its own within the first week, before commencing control measures. In the absence of control measures, only simulations predicting this initial rapid die-out showed elimination. Our analysis builds on the epidemiological modeling to incorporate and evaluate the benefits and costs of preventing raccoon rabies spread in the province.

 

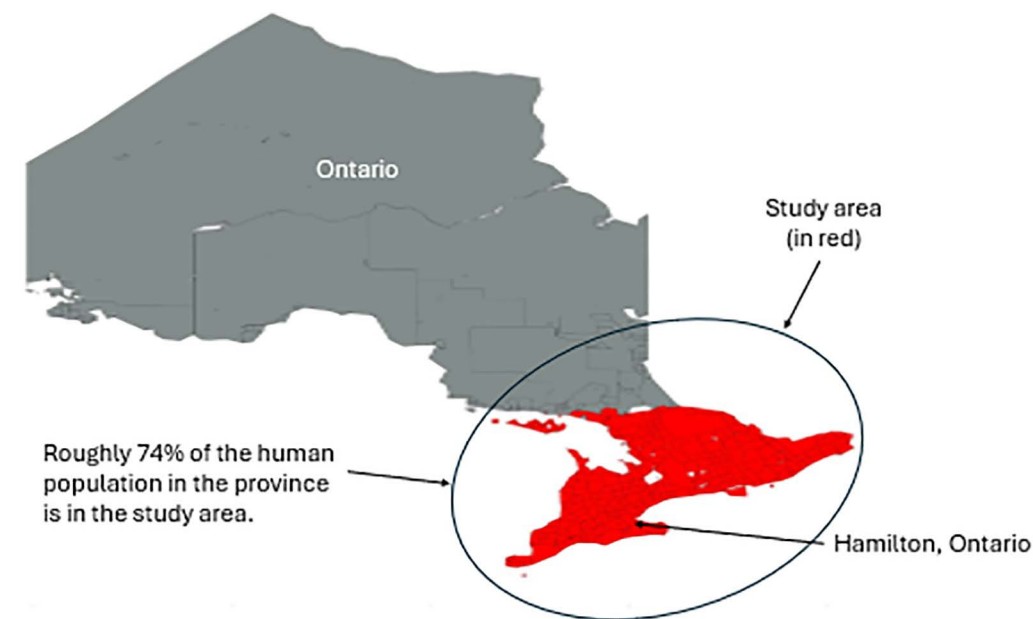

**Fig 1. Map of Ontario with the study area shown in red.** The first detected case was in Hamilton. Percent of the overall human population accounted for by the study area is highlighted. The base layer for the map was created using Canadian census sub division shapefiles obtained using the canadianmaps R package. The R package uses shapefile data provided by Statistics Canada. https://www12.statcan.gc.ca/census-recensement/2021/geo/sip-pis/boundary-limites/index2021-eng.cfm?year=21. https://www.statcan.gc.ca/en/reference/licence.

The primary costs associated with raccoon rabies include human PEP, animal rabies tests (AT), human exposure investigations (INVT), livestock deaths, pet vaccinations, and public education [3,7,11,12]. To quantify the benefits of successful intervention, this study compares a hypothetical intervention scenario with a no-intervention scenario and similar interventions in other North American regions to estimate the number of individuals protected or "saved."

## Methods

The results of the epidemiological disease spread model demonstrate that raccoon rabies was most likely to be eliminated when control interventions were in place until 2025 and least likely to be eliminated when no control interventions were used at all (Fig 2). In the intervention scenario, 67.4% of simulations predicted that rabies would be eliminated from the Southern Ontario region by 2025. For all the scenarios, 18.2% of simulations predicted rabies would die out one week after its introduction when controls had not yet been applied. In the scenario where no control interventions were used, the only simulations that predicted rabies elimination were those where rabies died out one week after its introduction.

This paper then utilizes the information from the agent-based epidemiological modeling of potential raccoon rabies spread illustrated in Fig 2, to determine the benefits and costs of stopping the spread of raccoon rabies in the province. Estimation of damages avoided, or benefits were calculated over the entire period since the initiation of the control program. To provide a present value of past benefits and costs, all estimates of costs and benefits were grown into 2023 Canadian dollars using the Canadian Consumer Price Index (CPI). The Canadian CPI data was publicly sourced from the Social Security Administration [30]. Based on publicly available data from the Bank of Canada, an average rate of inflation for the study period of 3% was utilized [31]. For this analysis we incorporated a 3% real discount rate into the estimation of benefits and costs accrued over the time period to compare benefits and costs over the lifetime of the project.

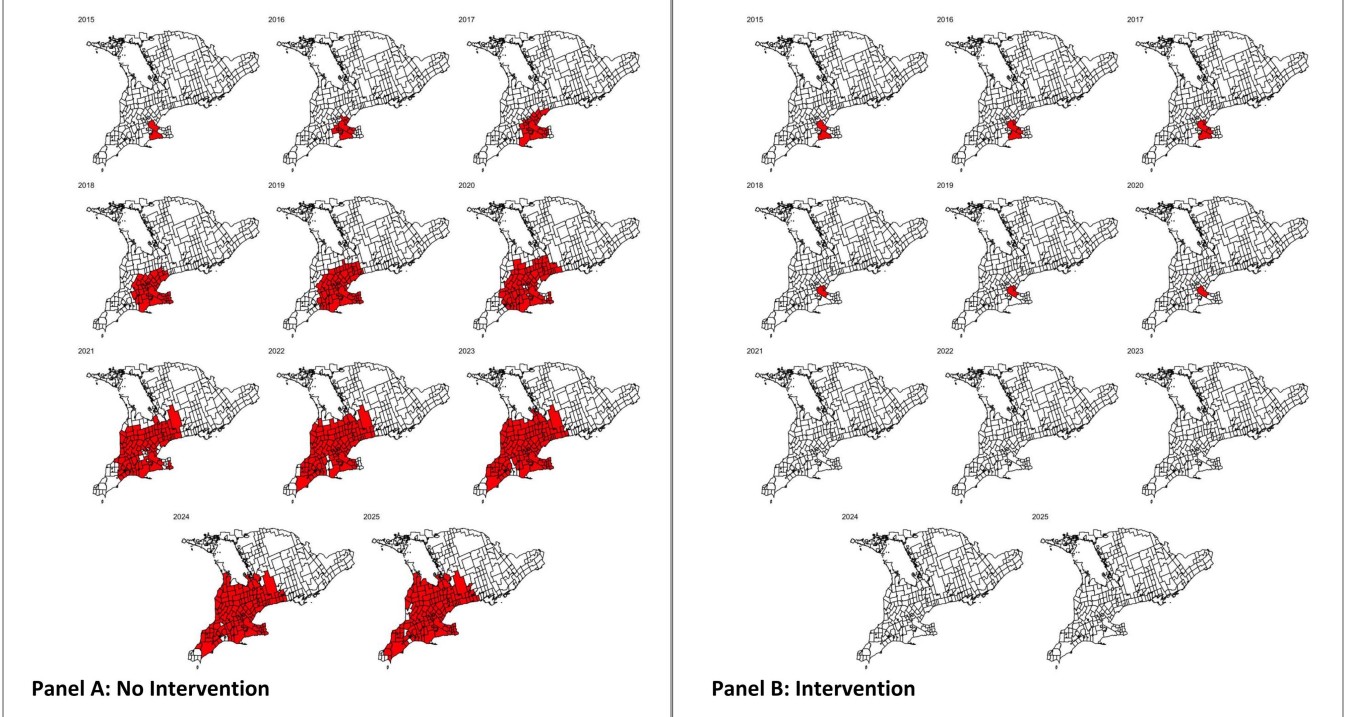

*Municipalities are colored red if, from the 1000 model simulations, the region had a median value of 1 or more infections in a year, 0 otherwise.

**Fig 2. Epidemiological disease spread maps of the study area from 2015-2025.** In Panel A, no intervention or control methods were employed, and disease spread is indicated in red. In Panel B, control was implemented in 2015 to 2025, and disease spread is indicated in red. Adapted from [23].

## Derivation of costs

A combination of actual and estimated costs was used to determine the total programmatic costs (TC) associated with the raccoon rabies control program. Programmatic costs were incurred for components related to the aerial and ground distribution of ORV baits, as well as for TVR, and enhanced surveillance. From 2015 to 2023, actual program costs were available and utilized for the study, and for 2024 and 2025 estimates were used due to the unavailability of actual costs.

## Derivation of benefits

Estimating the economic benefits of preventing the spread of raccoon variant rabies into Ontario, and potentially the rest of Eastern Canada, requires the monetization of damages avoided, which uses the value of resources protected as a measure of the benefits. The benefits of the control program were calculated as the savings from reducing the number of PEP, AT, and INVT necessary, plus the associated costs that would potentially be borne by individuals because of rabies exposure. These avoided costs make up most benefits derived from rabies control programs [3,9,10,15,16,32,33].

   To calculate benefits, the human population at risk (HPR) under each scenario was first determined. The epidemiological disease spread model from Acheson et al., (2023) predicted the spatial spread of the raccoon rabies variant through the province under each scenario (Fig 2). Census subdivision-level population data for the southern Ontario region was collected from Statistics Canada [34]. This data spans the period 2001–2021. The study period, however, spans the period 2015–2025. Since population forecasts do not exist at the census subdivision-level, we conducted a non-seasonal Autoregressive Integrated Moving Average (ARIMA) modeling exercise for each census subdivision in the study to obtain population estimates for years 2022–2025.

During the study period the average annual number of people in Ontario was approximately 14.88 million. In 2015 there were approximately 13.79 million and an estimated 16.18 million in 2025 [34]. This analysis did not incorporate the entire population of Ontario for the HPR and only utilized the municipalities identified in the total study area presented in Figs 1 and 2. All benefits rely on the calculation of the number of individuals protected by the program, or the human population saved (HPS). The difference between the estimated HPR with and without intervention provides the HPS and we used that to determine the number of PEP treatments per 100,000 people and combined that with AT and INVT rates sourced from previously published studies [3].

$$C_\alpha^i = (PEP + AT + INVT) * HPS^i$$

(1)

Where i is the year and α relates to low, medium, or high values of each of the variables. Annual total benefits (TB) consisted of the summation of the cost savings (C) related to reduced PEP, AT, and INVT, at each level α (low, medium, high) scaled by the HPS for the same year. To calculate the TB over the entire study period (2015–2025), for a certain level of α is represented by Equation 2.

$$TB_\alpha = \sum_{i=2015}^{2025} C_\alpha$$

(2)

Data were sourced on the range of annual case frequency of PEP and AT during raccoon rabies epizootics in the Eastern United States and Eastern Canada as proxies [3]. Data regarding annual PEP and AT rates reported in New Jersey [10], New York [32], and New Brunswick [3] were used to determine the hypothetical case frequency range that could have existed in the absence of a raccoon rabies control program (no intervention). New Jersey (NJ), New York (NY), and New Brunswick (NB) raccoon rabies epizootic PEP rates were reported as 66, 43.5, and 14 per 100,000 people, respectively, while AT rates were reported as 483, 65 and 45 per 100,000 people.

Determination of the number of INVT was accomplished using data the same methodology provided in Shwiff et al., (2008) that extrapolated data from Quebec, from 1995 to 2006 (excluding bat exposure) that indicated that for every PEP administered a range, 4.07–20.17 INVT occurred. Therefore, for this study, the number of INVT was directly estimated based upon the PEP per 100,000. The total number of PEP, AT, and INVT were summed for each year across each level of α. For example, when α=low, the frequency rates of all of variables were at their low, or NB, values and INVT measured 4 per 100,000 and when α=high, the frequency rates of all the variables were at their high, or NJ, values and INVT measured 20 per 100,000.

The predicted frequency levels of each of the variables were then combined with the cost information for each of the variables. PEP costs were a combination of direct and indirect costs associated with rabies exposure. Direct costs refer to the vaccine, other biologics, and the health professional salaries, while indirect costs refer to over-the-counter medicines, travel to physicians, and lost time from work associated with human rabies exposure. Indirect costs composed approximately one-third (33%) of the total costs associated with a rabid animal exposure [3,35]. Ontario PEP costs used for this analysis were $2,793 CAD including indirect costs ($2,100 direct + $693 indirect) (H. McClinchy, personal communication, December 1, 2023). Cost for AT was estimated at $300 CAD and INVT was $200 CAD for the analysis [3]. The number of individuals protected or 'saved' by the program was then combined with the rates of incidence of PEP, AT and INVT per 100,000 people to monetize the benefits of the program. Therefore, the TB calculated represents the present value of the entire raccoon rabies control program for each year, summed from 2015 to 2025 and grown into 2023 CAD.

### Benefit-cost ratios

Standard benefit-cost ratios were calculated for all α levels, low, medium, and high (NB, NY, and NJ) of human PEP treatments, AT and INVT for rabies that were estimated for the epizootic area. A ratio of 1.0 indicated that the benefits

PLOS Neglected Tropical Diseases

(savings) and costs were equal, or in other words, 1 unit of costs yielded 1 unit of benefits. A benefit-cost ratio > 1.0 indicated that the benefits of the program outweighed the costs and that the monies allocated were economically efficient. In programs that span several years, many times the best determination of efficiency is over the entire lifetime of the project rather than over 1 year, therefore the ratios used compared the benefits and costs (in 2023 CAD) of the entire program as opposed to on an annual basis. The costs of the entire program were compared to the total benefits that would have accrued from 2015 through 2025.

The benefit-cost ratios (BCR) associated with the epizootic area were determined by use of Equation 3:

$$BCR_\infty = TB_\alpha / TC$$

(3)

## Results

### Human population at risk

During the study period, the average annual number of people in Ontario was approximately 14.88 million. In 2015 there were approximately 13.79 million and an estimated 16.18 million in 2025. For this analysis, we only utilized the municipalities identified in the total study area presented in Figs 1 and 2 to calculate HPR and therefor make our calculation of HPS. The estimation of the HPR for the no intervention scenario provides the most expansive estimation of individuals potentially at risk of rabies exposure. Fig 3 provides the HPR with and without intervention and the estimate of HPS.

The total benefits calculated represent the value of the entire raccoon rabies control program for each year, in 2023 CAD, from 2015 to 2025. Total benefits accruing to the raccoon rabies control program were the calculated savings owing to the program (the projected prevented number of PEPs, ATs and INVTs under the intervention scenario) over the study period 2015–2025 (Table 1).

When α = low, the cost savings generated result from the lowest level of each variable (Table 1). Under these conditions, the intervention scenario provides overall program benefits of over $53 million CAD, mainly owing to saving related

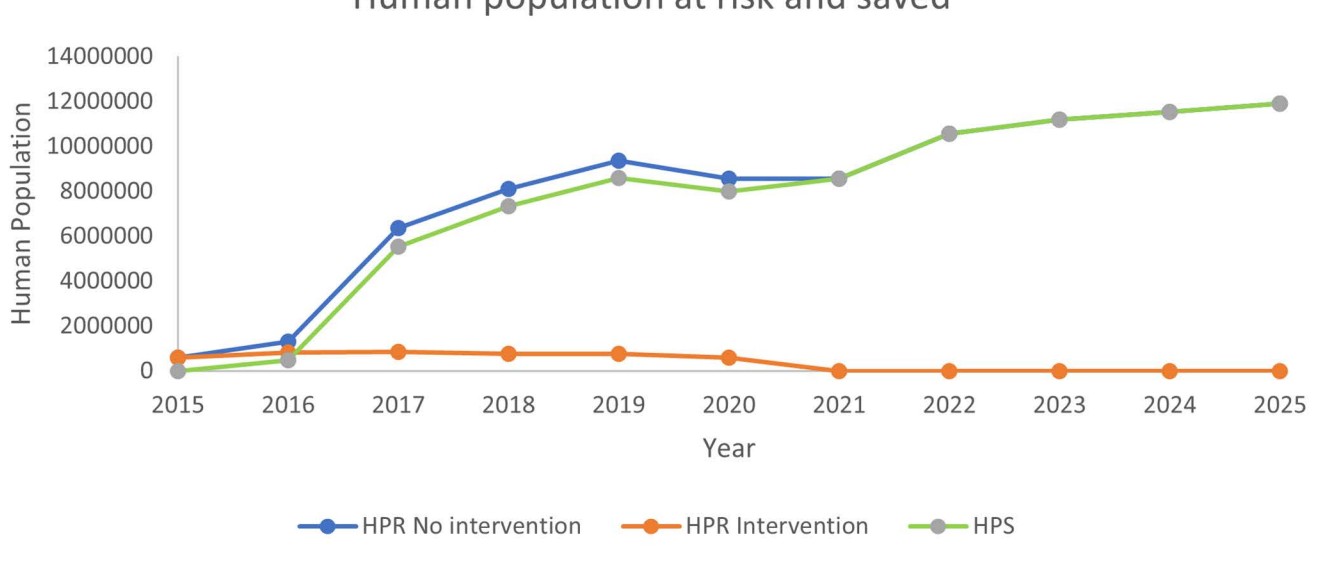

**Fig 3. Human population at risk without intervention and because of the projected spread of raccoon rabies and human population saved with raccoon rabies intervention.**

Table 1. Cost savings associated with raccoon rabies prevention in Ontario 2015-2025 at three levels of α*.

| Year | HPS | PEP (α=low) | AT (α=low) | INVT (α=low) | Total Savings (α=low) | PEP (α=med) | AT (α=med) | INVT (α=med) | Total Savings (α=med) | PEP (α=high) | AT (α=high) | INVT (α=high) | Total Savings (α=high) |
|---|---|---|---|---|---|---|---|---|---|---|---|---|---|
| 2015 | – | 0 | 0 | 0 | 0 | 0 | 0 | 0 | 0 | 0 | 0 | 0 | 0 |
| 2016 | 482,927 | 108,834 | 65,195 | 54,088 | 308,117 | 586,735 | 94,171 | 504,176 | 1,185,081 | 280,218 | 695,415 | 1,274,927 | 2,260,560 |
| 2017 | 5,521,343 | 2,158,956 | 745,381 | 618,390 | 3,522,727 | 6,708,183 | 1,076,662 | 5,768,282 | 13,549,127 | 10,177,993 | 7,950,734 | 14,576,346 | 32,705,813 |
| 2018 | 7,330,428 | 2,866,344 | 988,608 | 821,008 | 4,676,960 | 8,906,140 | 1,429,433 | 7,652,967 | 17,988,540 | 13,512,764 | 10,555,816 | 19,352,330 | 43,420,911 |
| 2019 | 8,582,144 | 3,355,790 | 1,158,589 | 961,200 | 5,475,580 | 10,426,919 | 1,679,518 | 8,959,758 | 21,060,195 | 15,820,153 | 12,358,287 | 22,656,360 | 50,835,800 |
| 2020 | 7,983,257 | 3,121,848 | 1,077,821 | 894,192 | 5,093,860 | 9,700,027 | 1,556,852 | 8,325,147 | 19,582,026 | 14,717,282 | 11,496,754 | 21,077,382 | 47,291,419 |
| 2021 | 8,568,675 | 3,350,523 | 1,156,771 | 959,692 | 5,466,986 | 10,410,555 | 1,670,892 | 8,945,697 | 21,027,143 | 15,795,324 | 12,338,282 | 22,621,882 | 50,755,488 |
| 2022 | 10,566,311 | 4,131,629 | 1,426,452 | 1,183,427 | 6,741,518 | 12,837,592 | 2,060,431 | 11,031,229 | 25,929,252 | 19,477,726 | 15,215,488 | 27,895,061 | 62,588,275 |
| 2023 | 11,193,169 | 4,376,753 | 1,511,078 | 1,253,635 | 7,141,466 | 13,599,197 | 2,183,668 | 11,685,668 | 27,467,533 | 20,683,264 | 16,118,163 | 29,549,866 | 66,351,303 |
| 2024 | 11,518,175 | 4,503,837 | 1,554,954 | 1,290,036 | 7,348,826 | 13,994,064 | 2,246,044 | 12,024,975 | 28,265,083 | 21,232,373 | 16,586,172 | 30,407,682 | 68,226,227 |
| 2025 | 11,512,264 | 4,657,933 | 1,608,156 | 1,334,174 | 7,600,263 | 14,472,865 | 2,322,951 | 12,436,404 | 29,232,160 | 21,958,829 | 17,153,660 | 31,448,377 | 70,560,866 |
| Total | 85,659,293 | 32,712,457 | 11,294,985 | 9,380,941 | 53,376,382 | 101,642,276 | 16,313,562 | 87,340,382 | 205,296,248 | 154,215,828 | 120,463,262 | 220,869,534 | 495,548,788 |

*In 2023 CAD.

to PEP and long-term impacts of not controlling rabies in the non-intervention scenario. When α = medium, the intervention scenario provides overall over $205 million CAD in program benefits, mainly owing to saving related to PEP and INVT. Lastly, when α = high, so intervention provides the greatest levels of damages avoided resulting in overall program benefits of over $495 million CAD, mainly owing to savings related to INVT. Resulting benefit-cost ratios indicated that at all levels of α, the program was economically efficient, returning $1.5 to $14.16 CAD in savings for every dollar expended on the intervention program.

## Discussion

Wildlife can play a crucial role in the emergence of zoonosis. Rabies is a deadly zoonosis, and this study sought to quantify the benefits and costs of a successful rabies elimination program, to highlight the economic efficiency of the program, and to provide the blueprint and impetus for other zoonotic disease management efforts. This study relied on a previously published epidemiological disease spread model to form the foundation for the economic estimate of limiting the spread of rabies. One main difficulty in modelling of most zoonotic diseases lies in the accurate prediction of HPR. Previously reported methodologies for the calculation of the HPR for rabies either considered the entire study area at risk once the first case has been detected [10,33] or considered that only a restricted part of the study area was at risk at the detection of the first case and then expand through the area based on arbitrary choices [8] or according to the expansion of the epizootic front at constant speed [36,37]. This study relied upon the simulation of disease spread within the raccoon population, as predicted by the epidemiological model in which at least one raccoon in the county was positive to determine the quantification of HPR. Results indicate that early elimination of the raccoon epizootic provides significant cost savings and should be prioritized as an objective of the program.

Some limitations exist with this study including determining the hypothetical annual frequencies of public health interventions (PEP, AT, and INVT) that would have existed in the absence of the raccoon rabies control program, which were used to calculate damages avoided or savings in the economic analysis. The estimated frequencies were based on information from the raccoon rabies epizootics in New Brunswick (low), New York (medium), and New Jersey (high). The use of these averages and adjusted frequencies reduces uncertainty of the monetary value of damages avoided; however, inclusion of additional information would further refine the results, thus underlining the need for more reported and published data on the effects of public health interventions rabies epizootics on.

It should be noted that not all benefits created by controlling raccoon rabies could be captured in this analysis. For example, the savings associated with a reduction in companion animal and livestock vaccinations along with any treatment (e.g., emergency care, quarantine, medications) were not factored into the analysis because of data limitations. Additionally, health benefits were not included in this analysis because of the lack of data necessary to make this estimate. Omission of these benefits creates more conservative program benefit estimate [22]. This analysis also included some factors that may have contributed to biased results including the incorporation of the total HPR in the study area in the calculation of HPS. Determination of the inclusion of a municipality's population was determined by the disease spread model and a municipality was included when simulations indicated that at least one positive raccoon was recorded within the municipality. As a result, it is possible that the total HPR is overstated which would in turn overstate the level of HPS, leading to an overestimate of programmatic benefits. Lastly, this analysis did not incorporate the potential impact of a reintroduction of the raccoon rabies variant over the study period. If a reintroduction had occurred and additional costs were incurred, this would negatively impact the benefits derived and positively impact the costs, leading to reduced BCRs and lower economic efficiency.

A range of potential program benefits (α = low, medium, and high) were estimated to compare with total actual and estimated intervention costs. This analysis of the control program to prevent the spread and eventual elimination of raccoon variant rabies in the province indicated significant saving for the modelled outcomes. Cost avoidance savings for a municipality may be accrued into perpetuity, or at least until the next rabies incursion occurs, however, those cost savings were not included in this analysis.

As the human population continues to expand at the human-wildlife-domestic livestock interface at increasing rates, the risk of disease transmission is heightened. Understanding the epidemiology and the benefits and costs of prevention of diseases like raccoon rabies, informs efficient intervention which plays a crucial role in safeguarding human health, protecting wildlife, and livestock. Additionally, this promotes a One Health approach to management that considers the interconnections between humans, animals, and the environment. Future research to model the economic impact of government spending to prevent the spread of raccoon rabies could utilize the methodology developed in this analysis to incorporate this information into epidemiological models to endogenously calculate costs and benefits of rabies programs to provide the most accurate understanding of the economics of disease elimination.

## Supporting information

**S1 Data. Raw data utilized to derive estimates of programmatic benefits.**
(XLSX)

## Author contributions

**Conceptualization:** Stephanie A. Shwiff, Emily S. Acheson, Patrick A. Leighton, Larissa Nituch, François Viard, Tore Buchanan.

**Data curation:** Stephanie A. Shwiff, Emily S. Acheson, Levi Altringer, Tore Buchanan.

**Formal analysis:** Stephanie A. Shwiff, Levi Altringer.

**Investigation:** Stephanie A. Shwiff, Sarah Sykora.

**Methodology:** Stephanie A. Shwiff.

**Project administration:** Stephanie A. Shwiff, Tore Buchanan.

**Supervision:** Stephanie A. Shwiff, Tore Buchanan.

**Writing – original draft:** Stephanie A. Shwiff, Levi Altringer.

**Writing – review & editing:** Stephanie A. Shwiff, Emily S. Acheson, Patrick A. Leighton, Larissa Nituch, Sarah Sykora, François Viard, Tore Buchanan.

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
