## [Decision Letter · Decision Letter 0]

18 Feb 2025

PNTD-D-24-01269

Benefit-Cost Analysis of Raccoon Rabies Control in Ontario, Canada

Dear Dr. Shwiff,

Thank you for submitting your manuscript to PLOS Neglected Tropical Diseases. After careful consideration, we feel that it has merit but does not fully meet PLOS Neglected Tropical Diseases's publication criteria as it currently stands. Therefore, we invite you to submit a revised version of the manuscript that addresses the points raised during the review process.

Please submit your revised manuscript within 60 days Apr 19 2025 11:59PM. If you will need more time than this to complete your revisions, please reply to this message or contact the journal office at plosntds@plos.org. Please include the following items when submitting your revised manuscript:

We look forward to receiving your revised manuscript.

Kind regards,

Abdallah Samy

Section Editor

Shaden Kamhawi

co-Editor-in-Chief

Paul Brindley

co-Editor-in-Chief

**Journal Requirements:**

At this stage, the following Authors/Authors require contributions: Stephanie A Shwiff, Emily Acheson, Levi Altringer, Patrick Leighton, Larissa Nituch, Sarah Sykora, François Viard, and Tore Buchanan. Please ensure that the full contributions of each author are acknowledged in the "Add/Edit/Remove Authors" section of our submission form.

Potential Copyright Issues:

- Figures 1 and 2. Please provide a direct link to the base layer of the map (i.e., the country or region border shape) and ensure this is also included in the figure legend; and provide a link to the terms of use / license information for the base layer image or shapefile. We cannot publish proprietary or copyrighted maps (e.g. Google Maps, Mapquest) and the terms of use for your map base layer must be compatible with our CC BY 4.0 license.

5) We note that your Data Availability Statement is currently as follows: "Data is available at National Wildlife Research Center's website". Please confirm at this time whether or not your submission contains all raw data required to replicate the results of your study. Authors must share the “minimal data set” for their submission. PLOS defines the minimal data set to consist of the data required to replicate all study findings reported in the article, as well as related metadata and methods (https://journals.plos.org/plosone/s/data-availability#loc-minimal-data-set-definition).

- The points extracted from images for analysis..

**Reviewers' Comments:**

Reviewer's Responses to Questions

**Key Review Criteria Required for Acceptance?**

**Methods** :

-Are the objectives of the study clearly articulated with a clear testable hypothesis stated?

-Is the study design appropriate to address the stated objectives?

-Is the population clearly described and appropriate for the hypothesis being tested?

-Is the sample size sufficient to ensure adequate power to address the hypothesis being tested?

-Were correct statistical analysis used to support conclusions?

-Are there concerns about ethical or regulatory requirements being met?

Reviewer #1: Yes indeed. Objective of the study was well articulated. The study design is also appropriate with sufficient sample size. the issue of ethical clearance was addressed and samples were analysed with proper statically identified software.

Reviewer #2: see general comments

Reviewer #3: The objective of the study is generally clearly articulated and the study design is appropriate.

The end of the final sentence of the intro , "and similar intervention in other areas of North America" is unclear and I think can be deleted.

In the intro, lines 135 to 145 on p6 may be better suited for the methods or results section.

More details are needed about the analysis, in particular the perspective and how discounting was implemented. It seems that discounting may not have been used despite the long time horizon of the study - if that's the case, a justification should be provided.

I would suggest putting all of the model parameters and assumptions in a table with reference for easy review.

The equations as currently written don't seem correct. In particular, "i" isn't defined, and as currently written, it seems to imply that costs are being summed across scenarios (alpha).

Some of the next in the methods could be moved to the discussion of limitations (e.g., lines 218-222 on page 11).

The exclusion of health benefits associated with rabies control is not mentioned (and is a departure from usual CBA methods). The authors should explain why this is the case (lack of data? small expected utility values?), as this will also likely underestimate the full economic value.

Results lines 260-264 on p. 13 should be moved to the methods.

An editorial suggestion: it may be more intuitive/logical for readers to put the methods for program costs ahead of the benefits.

Reviewer #4: This paper reports on an economic analysis of raccoon rabies control in Canada. This is a very useful exercise as we are lacking evidence for the profitability of One Health interventions. However, there are a series of edits and questions to this manuscript, that should be addressed:

It is not clear how the Human Population Risk was validated. This is a limitation of the study that should be stated.

If I did not overlook it, there is no discounting considered in the BCA. As this is a multi-year exercise, The annual benefits and costs should be discounted with an indicated rate and finally net present values of benefits and cost should be related

**Results** :

-Does the analysis presented match the analysis plan?

-Are the results clearly and completely presented?

-Are the figures (Tables, Images) of sufficient quality for clarity?

Reviewer #1: The results were well presented. Figures and tables were clearly illustrated.

Reviewer #2: see general comments

Reviewer #3: The data in Fig 3 are also contained in the tables, so could consider removing the figure.

For Tables 1-3, I think the Incidence and Cost columns represent incidence and costs averted with intervention. If so, consider relabelling to make this clear.

Reviewer #4: Figure 3: Turn the text of y-axise to read from bottom to top. It is not clear if the intervention-cost depicted in Figure 3 are empirical or simply model based? I looks like the intervention has an immediate effect on human risk which is rare for mass vaccinations. Hence, the authors should state about the empirical validity of their simulations. In wildlife, re-introductions from outside the intervention area are common. Did the authors consider re-introduction in their model or was this not included. This should be stated.

**Conclusions** :

-Are the conclusions supported by the data presented?

-Are the limitations of analysis clearly described?

-Do the authors discuss how these data can be helpful to advance our understanding of the topic under study?

-Is public health relevance addressed?

Reviewer #1: Conclusions were well summarized using the data and the discussions of authors were helpful for other authors and public health concerns.

Reviewer #2: see general comments

Reviewer #3: Would be useful to expand the discussion about the costs and benefits associated with raccoon variant rabies elimination. The authors mention that the cost avoidance savings accrue into perpetuity, but are there also ongoing costs associated with maintaining elimination?

Although a range of scenarios are presented, can the authors provide some commentary on what scenarios are considered more/or less probable? i.e., are all considered equally probable or is one more probable?

Reviewer #4: Finally, the authors should refer to comparable studies and new methods, using for example game-theoretical strategy analysis and human capital benefits [1].

1. Bucher, A., Dimov, A., Fink, G., Chitnis, N., Bonfoh, B., Zinsstag, J., Benefit-cost analysis of coordinated strategies for control of rabies in Africa. Nature Communications, 2023. 14: p. 5370.

**Editorial and Data Presentation Modifications?**

Reviewer #1: No comment

Reviewer #2: see general comments

Reviewer #3: The abstract includes information that is not included in the rest of the manuscript (scenario 2) and focus more on summarizing the epidemiological modelling that informs the CBA. The abstract should be revised to focus on the methods used in the current study.

Reviewer #4: I made suggestions for edits of the tables and figures

**Summary and General Comments** :

Reviewer #1: The cost benefit analysis study topic has significant contribution as more studies and publications have not clearly work on cost benefit analysis of a given subject of study.

Reviewer #2: General comments:

Applying economics to support decision-making in the fight against animal diseases is becoming increasingly important. This applies in particular to the control of wildlife diseases, where the benefits of the measures are often questioned. The paper by Shwiff et al. deals with a benefit-cost analysis to examine the economic benefits of intervention strategies in the frame Canada’s largest raccoon rabies outbreak in history which began in 2015.

To this end they used an epidmiological disease spread model and compared three different intervention strategies with scenario 1 and 2 estimating the spread of the raccoon rabies variant if control measures were continued until 2025 or ended in 2020, respectively, and scenario 3 as a no-intervention strategy. Results from this analysis indicated savings for Ontario ranging from $53 million to $495 million CAD over the study period (2015 to 2025) compared to a no-intervention strategy depending on low, medium, or high estimates of each of the variables.

This is interesting information that is worth publishing, however, one cannot help but get the impression that the paper was written in a rush. The paper sometimes lacks the diligence required for publication, but this can be easily remedied. To be clear, there is no doubt about the methodology and modeling used, it is rather the uninspiring, mathematically undercooled presentation. One would have liked a more interesting and vivid presentation of the manuscript for the reader. A little bit more of information and input would give the reader a better picture.

Here some points to consider for improving the manuscript.

• The introduction section is less inspiring and informative. A brief introduction to the topic (the first author herself wrote an nice article on the topic - Shwiff et al., 2018) and mentioning of published literature on rabies specific cost-benefit analyses would be desirable. It would also be good to briefly describe what concrete measures have been taken since 2015 to contain raccoon rabies by referring to published literature instead of focusing extensively on the agent-based model and roles and responsibilities of different players.

• Often references are missing and should be provided whenever needed (see lines 111, 113, 178, .

• Lines 182-189: This part is a little confusing. Can the AT and INVT rates from a area endemic for dogs rabies be compared one-to-one with those in an area endemic for raccoon rabies (see reference 17)? These are two different scenarios. In the former the biting incidence per 100,000 inhabitants and the resulting PEP can be much higher. Two sentences later it is stated that data were sourced on the range of annual case frequency of PEP and AT during raccoon rabies epizootics in the Eastern United States and Eastern Canada as proxies. How does this fit with the information provided above?

• Lines 217-222: Would this part not better suited for the Discussion section?

• Line 227: We are now in the year 2025, so “forecasted” sounds a bit strange. The costs for 2023 and 2024 should actually be available and for this year we should know what is to be spent. Would it be possible to provide a annual breakdown of costs?

• Line 229: What costs are included here, e.g. costs for TVR, ORV (vaccine price, costs for bait distribution, transport, storage…) etc.. Is it direct costs only?

• Line 250: There is no mention of the total population of Ontario and in particular the population living in the area shown in Figures 1 and 2. The HRP appears to be quite high. From a practical perspective, it is hard to imagine that 75% of the population in Ontario would be at risk in a no-intervention scenario.... Could it be that the modeling assumptions are a bit biased? Could this be discussed under the limitations of the study?

• Lines 260-264: This belongs to Materials & Methods.

• Lines 270-273, 275-277, 279-281: For the three tables always the same text wording is used.

• Discussion: This part needs improvements. The significance, importance and relevance of the results obtained should be objectively discussed against available literature on the topic (Meltzer, 1996; Elser et al., 2016; Bucher et al., 2023). The limitations of the study should more clearly discussed (see above).

• Figures: the captions are generally very poor.

• Figure 1 and 2: These do not appear to be original figures. Please provide the source or describe how they were made. Also, only the southeasternmost part of Ontario is shown in the figures. Since not everyone is familiar with the topography, a map of Ontario with an enlargement of the area under study would be desirable so that the unfamiliar reader can orient himself better.

• Figure 3: The y-achsis should be adjusted to zero.

• Tables 1-3: There is redundant information. The figures in the first three columns of each table are already shown graphically in Figure 3 and can be removed. When the number of population saved (column 4) could be included in Figure 3 as well, than all three tables could be combined in one only showing the outcome if different levels of benefit are applied.

• Table 4 is redundant as it is just a summary of Tables 1-3.

Reviewer #3: This is a valuable study that assesses the cost-benefit of raccoons rabies control in Ontario, Canada. Quantifying the value of prevention is important for ensuring ongoing support for programs such as this. Strengths of this analysis include the use a previously published epidemiologic model to develop a "no intervention" counterfactual scenario and consideration of a range of outcomes using data from other jurisdictions. Overall, the study would be strengthened by more details about the economic analysis.

Reviewer #4: This paper merits publication if the methodological questions, mainly of the discounting are resolved.

PLOS authors have the option to publish the peer review history of their article (what does this mean? ). If published, this will include your full peer review and any attached files.

**Do you want your identity to be public for this peer review?** For information about this choice, including consent withdrawal, please see our Privacy Policy .

Reviewer #1: No

Reviewer #2: No

Reviewer #3: No

Reviewer #4: No

**Figure resubmission:**

**Reproducibility:**



---

## [Decision Letter · Decision Letter 1]

9 Aug 2025

Benefit-cost analysis of raccoon rabies control in Ontario, Canada

Dear Dr. Shwiff,

Thank you for submitting your manuscript to PLOS Neglected Tropical Diseases. After careful consideration, we feel that it has merit but does not fully meet PLOS Neglected Tropical Diseases's publication criteria as it currently stands. Therefore, we invite you to submit a revised version of the manuscript that addresses the points raised during the review process.

Please submit your revised manuscript within 60 days Sep 08 2025 11:59PM. If you will need more time than this to complete your revisions, please reply to this message or contact the journal office at plosntds@plos.org. Please include the following items when submitting your revised manuscript:

We look forward to receiving your revised manuscript.

Kind regards,

Abdallah M. Samy, PhD

Section Editor

Abdallah Samy

Section Editor

Shaden Kamhawi

co-Editor-in-Chief

Paul Brindley

co-Editor-in-Chief

**Journal Requirements:**

1) 

Some material included in your submission may be copyrighted. According to PLOS’s copyright policy, authors who use figures or other material (e.g., graphics, clipart, maps) from another author or copyright holder must demonstrate or obtain permission to publish this material under the Creative Commons Attribution 4.0 International (CC BY 4.0) License used by PLOS journals. Please closely review the details of PLOS’s copyright requirements here: PLOS Licenses and Copyright. If you need to request permissions from a copyright holder, you may use PLOS's Copyright Content Permission form.

Potential Copyright Issues:

i) Figure 1: please provide a direct link to the base layer of the map (i.e., the country or region border shape) and ensure this is also included in the figure legend.

**Reviewers' Comments:**

Reviewer's Responses to Questions

**Key Review Criteria Required for Acceptance?**

**Methods**

-Are the objectives of the study clearly articulated with a clear testable hypothesis stated?

-Is the study design appropriate to address the stated objectives?

-Is the population clearly described and appropriate for the hypothesis being tested?

-Is the sample size sufficient to ensure adequate power to address the hypothesis being tested?

-Were correct statistical analysis used to support conclusions?

-Are there concerns about ethical or regulatory requirements being met?

Reviewer #2: Yes, all of the reviewers' comments were addressed appropriately.

Reviewer #3: (No Response)

Reviewer #4: In there letter of revision the authors claim that the changes in the revised manuscript are in blue. But I don’t see any such changes. I am afraid that they uploaded the wrong manuscript. I can not judge the reivsions if I don’t see the revised manuscript.

I would also appreciate if the authors respond to the query in the letter of revision and not only mention that a change is made in the text.

**Results**

-Does the analysis presented match the analysis plan?

-Are the results clearly and completely presented?

-Are the figures (Tables, Images) of sufficient quality for clarity?

Reviewer #2: Yes, all of the reviewers' comments were addressed appropriately.

Reviewer #3: I may have missed it, but I don't see the program costs presented in the results, just the savings.

Reviewer #4: (No Response)

**Conclusions**

-Are the conclusions supported by the data presented?

-Are the limitations of analysis clearly described?

-Do the authors discuss how these data can be helpful to advance our understanding of the topic under study?

-Is public health relevance addressed?

Reviewer #2: Yes, all of the reviewers' comments were addressed appropriately.

Reviewer #3: (No Response)

Reviewer #4: (No Response)

**Editorial and Data Presentation Modifications?**

Reviewer #2: Accept

Reviewer #3: For the intro, I found the paragraph on p. 6 (lines 122 to 133) hard to follow in terms of the logic. The emphasis on gaps in high-risk, low-income settings seems less directly relevant to this study on raccoon rabies in a high-income context. I also wasn't sure how the mention of game theoretic methods fits in.

Reviewer #4: (No Response)

**Summary and General Comments**

Reviewer #2: The time the authors have spent responding to the reviewers' comments has definitely paid off. The quality of the manuscript has improved significantly; the work is now worthy of publication.

Reviewer #3: It is difficult to assess whether the specific reviewer comments were addressed, as the authors did not provide a response document mapping their revisions to the reviewer feedback. Additionally, the tracked changes version is not helpful for this purpose, since it shows the entire text as replaced, making it unclear where specific changes were made. I also note that the query regarding the use of discounting does not appear to have been addressed.

Reviewer #4: (No Response)

PLOS authors have the option to publish the peer review history of their article (what does this mean? ). If published, this will include your full peer review and any attached files.

**Do you want your identity to be public for this peer review?** For information about this choice, including consent withdrawal, please see our Privacy Policy .

Reviewer #2: No

Reviewer #3: No

Reviewer #4: No

**Figure resubmission:**

**Reproducibility:**



---

## [Decision Letter · Decision Letter 2]

29 Oct 2025

Dear Dr. Shwiff,

We are pleased to inform you that your manuscript 'Benefit-cost analysis of raccoon rabies control in Ontario, Canada' has been provisionally accepted for publication in PLOS Neglected Tropical Diseases.

Best regards,

Shaden Kamhawi

co-Editor-in-Chief

Paul Brindley

co-Editor-in-Chief

Reviewer's Responses to Questions

**Key Review Criteria Required for Acceptance?**

**Methods**

-Are the objectives of the study clearly articulated with a clear testable hypothesis stated?

-Is the study design appropriate to address the stated objectives?

-Is the population clearly described and appropriate for the hypothesis being tested?

-Is the sample size sufficient to ensure adequate power to address the hypothesis being tested?

-Were correct statistical analysis used to support conclusions?

-Are there concerns about ethical or regulatory requirements being met?

Reviewer #2: The methods have been appropriately revised.

**Results**

-Does the analysis presented match the analysis plan?

-Are the results clearly and completely presented?

-Are the figures (Tables, Images) of sufficient quality for clarity?

Reviewer #2: Results are now clearly presented.

**Conclusions**

-Are the conclusions supported by the data presented?

-Are the limitations of analysis clearly described?

-Do the authors discuss how these data can be helpful to advance our understanding of the topic under study?

-Is public health relevance addressed?

Reviewer #2: The revised conclusions support the data presented and also adequately address limitations.

**Editorial and Data Presentation Modifications?**

Reviewer #2: none

**Summary and General Comments**

Reviewer #2: This is a very thorough revision. All concerns have been adequately addressed or, where this was not possible, discussed. I have no further reservations.

PLOS authors have the option to publish the peer review history of their article (what does this mean? ). If published, this will include your full peer review and any attached files.

**Do you want your identity to be public for this peer review?** For information about this choice, including consent withdrawal, please see our Privacy Policy .

Reviewer #2: No

---

## [Editor Report · Acceptance letter]

Dear Dr. Shwiff,

We are delighted to inform you that your manuscript, "Benefit-cost analysis of raccoon rabies control in Ontario, Canada," has been formally accepted for publication in PLOS Neglected Tropical Diseases.

Best regards,

Shaden Kamhawi

co-Editor-in-Chief

Paul Brindley

co-Editor-in-Chief
